# Improving experience of medical abortion at home in a changing therapeutic, technological and regulatory landscape: a realist review

Paula Baraitser,[1] Caroline Free,[2] Wendy V Norman [iD] ,[3,4] Maria Lewandowska [iD] ,[4] Rebecca Meiksin,[4] Melissa J Palmer,[4] Rachel Scott [iD] ,[4] Rebecca French,[4] Kaye Wellings,[4] Alice Ivory,[2] Geoff Wong,[5] The SACHA study team

[1]King's College Hospital, NHS Foundation Trust, London, UK
[2]Faculty of Epidemiology and Population Health, London School of Hygiene & Tropical Medicine, London, UK
[3]Department of Family Practice, The University of British Columbia, Vancouver, British Columbia, Canada
[4]Faculty of Public Health and Policy, London School of Hygiene & Tropical Medicine, London, UK
[5]Nuffield Department of Primary Care Health Sciences, Oxford University, Oxford, UK

**Correspondence to**
Maria Lewandowska;
maria.lewandowska@lshtm.ac.uk

## ABSTRACT

**Objective** To inform UK service development to support medical abortion at home, appropriate for person and context.

**Design** Realist review

**Setting/participants** Peer-reviewed literature from 1 January 2000 to 9 December 2021, describing interventions or models of home abortion care. Participants included people seeking or having had an abortion.

**Interventions** Interventions and new models of abortion care relevant to the UK.

**Outcome measures** Causal explanations, in the form of context-mechanism-outcome configurations, to test and develop our realist programme theory.

**Results** We identified 12 401 abstracts, selecting 944 for full text assessment. Our final review included 50 papers. Medical abortion at home is safe, effective and acceptable to most, but clinical pathways and user experience are variable and a minority would not choose this method again. Having a choice of abortion location remains essential, as some people are unable to have a medical abortion at home. Choice of place of abortion (home or clinical setting) was influenced by service factors (appointment number, timing and wait-times), personal responsibilities (caring/work commitments), geography (travel time/ distance), relationships (need for secrecy) and desire for awareness/involvement in the process. We found experiences could be improved by offering: an option for self-referral through a telemedicine consultation, realistic information on a range of experiences, opportunities to personalise the process, improved pain relief, and choice of when and how to discuss contraception.

**Conclusions** Acknowledging the work done by patients when moving medical abortion care from clinic to home is important. Patients may benefit from support to: prepare a space, manage privacy and work/caring obligations, decide when/how to take medications, understand what is normal, assess experience and decide when and how to ask for help. The transition of this complex intervention when delivered outside healthcare environments could be supported by strategies that reduce surprise or anxiety, enabling preparation and a sense of control.

## STRENGTHS AND LIMITATIONS OF THIS STUDY

⇒ We present a systematic and transparent approach to the realist review, which we conducted in accordance with the RAMESES standards.
⇒ A broad range of content expertise from our authorship team, patients and the public informed the review.
⇒ Our analysis was conducted only on publicly accessible literature, located through recognised research databases and Google.
⇒ We found gaps in the evidence and have highlighted these in our conclusions.
⇒ Our findings are intended to be relevant to settings where abortion is legal, and primarily where it is delivered as part of a universal healthcare system.

## BACKGROUND

Changes in the therapeutic, technological and regulatory landscape are impacting abortion care and changing access to medical abortion at home. Medical abortion requires two medications, mifepristone (a competitive progestogen receptor antagonist) and the synthetic prostaglandin, misoprostol. Since mifepristone was first licenced for use in Great Britain in 1991, the proportion of all abortions that are medical abortions has increased steadily with a more recent shift towards medical abortion at home, where both the mifepristone and misoprostol or the misoprostol alone are taken at home. During the COVID-19 pandemic, these trends were accelerated and in 2020, 85% of all abortions completed in England and Wales, and 88% of those in Scotland, were medical abortions at home.[1 2] These developments reflect similar changes in Australia, Canada and Europe.[3–7]

Medical abortion at home is safe, effective and acceptable and clinical guidance on some elements, such as doses of abortion medications, are consistent.[7–21] However, significant

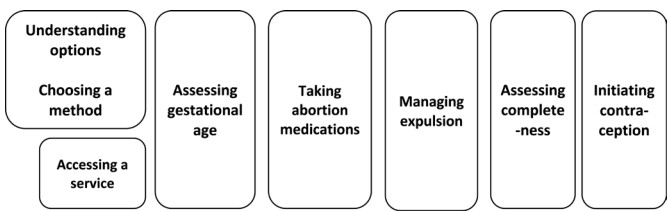

**Figure 1** Stages of the medical abortion process (initial theory of change).

variation in other elements of clinical pathways remains as providers test new approaches to providing information, support, pain relief and follow-up.[22] It is important to note that a minority (17%–34%) of people who participate in research on medical abortion at home would not choose this method if having an abortion again.[12 23–25] We are interested in the evidence to inform optimal configuration of medical abortion services at home and particularly in the differences in experience between populations and contexts.

### Terminology and settings
Our findings relate to settings where abortion is legal and where there is a choice of abortion method and setting. Recommendations may be different in settings where abortion is illegal, where mifepristone is not available, where medication combinations or doses are used that are not evidence based or where surgical options are not available or not safe. Women and people with other gender identities may have abortions. We have used the terms 'people who have an abortion' or 'people' to describe our population of interest.

### Objectives
► To synthesise evidence on user experience of medical abortion at home.
► To develop a realist programme theory to explain what interventions improve user experience for whom and in what context.
► To use this programme theory to develop recommendations for service providers and those having medical abortions at home.

### Review question
► What support, for whom and in what circumstances could improve the experience of people who have medical abortion at home?

### METHODS
The SACHA: *Shaping Abortion for Change* study brings together over 20 researchers from seven countries to create an evidence base to inform health service configuration for abortion provision in the UK (https://www.lshtm.ac.uk/research/centres-projects-groups/sacha#welcome). This review is part of this programme of work and draws on the expertise of the team.

In the literature review component of the study, we sought to investigate novel models of abortion care addressing current therapeutic, technological and

regulatory trends, which would be relevant to the UK in the next 5 years. We subsequently divided this work between two reviews: a scoping review discussing the healthcare practitioner and system preparedness for abortion provision, which is currently in progress and this realist review, focused on improving the experience of medical abortion at home.

We conceptualised medical abortion as a complex process that has context-sensitive outcomes which may be variably delivered and experienced. A realist approach enabled us to both account for and explain the influence of context on this variation.[26] Our protocol was registered with Prospero https://www.crd.york.ac.uk/prospero/display_record.php?RecordID=225307 and we followed the steps which are commonly used in realist reviews[27 28]—location of existing theories, evidence search and document selection, data extraction, data synthesis and development of a refined programme theory.

We used the PRISMA (The Preferred Reporting Items for Systematic Reviews and Meta-Analysis) checklist when writing our report.[29]

### Location of existing theories
A realist approach to understanding interventions (in our case home medical abortion) proposes that any intervention is underpinned by one or more theories (that may be implicit or explicit) in the design of the intervention. This theory sets out how the designers anticipate their intervention will work. In a realist review, this understanding is captured via an initial programme theory. Our initial programme theory was developed with input from the SACHA team. We charted the stages of home medical abortion from a user perspective to understand what type of support might be required from the healthcare system at each stage (figure 1). These stages within a user journey were used to structure our analysis.

### Searching for evidence and document selection
Our evidence search strategy started with a comprehensive literature search designed to answer a broader question for the SACHA Study: 'Interventions of models of abortion care/provision: addressing current therapeutic, technological and regulatory trends which would be relevant to or feasible in the UK in the next 5 years.' This search was designed to retrieve papers that would likely be relevant to both this realist review and a scoping review. The databases and search terms for this search are listed in online supplemental appendix 1 and the inclusion and exclusion criteria are listed in table 1. The search dates were 1 January 2000–9 December 2022.

Search results were imported into an electronic reference manager Covidence (https://www.covidence.org/) and screened in two stages. The first screen was based on title and abstract with a sample double-screened to check for consistency and discussion of studies of uncertain eligibility to reach consensus. The remainder of papers at the abstract and title stage and all those at the full text stage were single screened. The full text of all included

**Table 1**  Inclusion/exclusion criteria for the review

|  | Included | Excluded |
|---|---|---|
| Interventions | Interventions of models of abortion care/provision: addressing current therapeutic, technological and regulatory trends which would be relevant to, or feasible in the UK in the next 5 years | Interventions aiming to legalise abortion, mitigate the effects of illegal abortion or address the financial aspects of abortion access. Interventions relevant to abortion care in unregulated/poorly regulated contexts. Pharmacological studies. |
| Populations | People seeking/having had an abortion, those accompanying someone through an abortion process, healthcare workers reporting experience of/attitudes towards abortion provision. |  |

documents was uploaded into a second electronic reference management system (EndNote).

We screened the papers using the WHO database (https://abortion-policies.srhr.org/) to determine the legality of abortion in each of the studied settings.

We were confident that this search was comprehensive and contained all relevant information that was needed to answer our realist review question. We subsequently rescreened this search for the purposes of the realist review as follows:

1. A subject expert (WVN) screened all full texts to identify documents that were specifically related to medical abortion at home and then grouped these into the stages outlined in our intitial programme theory.
2. PB read these papers to identify whether they were relevant to developing our initial programme theory.

Papers were considered useful for theory development and testing where they offered empirical evidence that could refine, refute or confirm developing emerging realist programme theory and inform the development of context-mechanism-outcome configurations (CMOCs) within it[28] (online supplemental appendix 2). This meant including all papers that were relevant to people's experience of each stage of the medical abortion at home journey and the support provided that might modify this experience. We used a broad definition of 'support' to include any intervention by the individual having the abortion, the healthcare system or friends and family intended to improve or maintain health or well-being before, during and after the abortion process.

Realist review processes are iterative, so as our CMO configurations developed we revisited the list of included papers for the scoping review to look for additional relevant materials. We also completed additional searches as required including handsearching reference lists and completing new searches, to seek out more relevant data.

### Data extraction

The final selected papers regarding each stage of abortion care were read and reread by PB and, for the post abortion contraception, by CF. Findings were summarised in spreadsheets which contained information on key relevant findings from each paper, grouped according to the programme theory. Based on their interpretations

of these findings, PB and CF developed CMOCs for each stage of the medical abortion process. No uniform data set was extracted from each paper, rather the data (verbatim sections of text) of each paper that were relevant to each emerging CMOCs were grouped together and iteratively used during the analysis process (see below) to develop CMOCs.

### Data analysis and synthesis

A realist logic of analysis uses data to produce causal explanations for outcomes that occur within a programme theory in the form of CMOCs. A CMOC is a proposition that explains what element of an initiative works, for whom and in what circumstances and is the primary way of reporting findings within a realist review. Within a CMOC, the causal claim being made is (in its simplest form), when a particular context is present, it 'triggers' or 'activates' a particular mechanism, which causes a particular outcome. Mechanisms are hidden causal processes that are context-sensitive and are usually inferred based on interpretations of the data. Data to inform our interpretation of the relationships between contexts, mechanisms and outcomes were sought within and across documents (eg, mechanisms inferred from one document helped explain the way contexts influenced outcomes in a different document). Synthesising data from different documents is often necessary to compile CMOCs, since not all parts of the configurations will be articulated in every document. During our analysis, we used interpretive cross-case comparison to understand and explain how and why reported outcomes have occurred.

### Refining programme theory

We iteratively refined our initial programme theory as the review progressed based on our interpretations of the data from the included papers. For each stage of the abortion at home process theory we sought to unpack what support is needed. Our final programme theory contains CMOCs that explain the outcomes achieved by the support provided, why it happens and in which contexts.

### Patient and public involvement

The whole SACHA Study, including the realist review, was consulted by an Advisory Group with PPI members, who

helped in developing the research questions and methodology. The development of the research question and outcome measures were entirely informed by patients' priorities. Patient representatives were not involved in the conduct, analysis or interpretation of the results. The results of the study will be disseminated to the public and to patients who were involved in other components of the study through online newsletters, press releases, informational videos posted online and through other activities informed by the consultations with the Advisory Group.

## RESULTS

Our searches identified 27982 potentially relevant abstracts for both reviews. After duplicates and papers with incorrect dates or excluded literature types were removed 12401 titles and abstracts were screened with 944 selected for full text assessment. At this stage, 590 studies were excluded which did not meet the inclusion criteria or were duplicates. The remaining full texts were screened for potential relevance to the realist review question and 244 studies were categorised by a content expert (WVN) against the inclusion criteria and the initial programme theory. Additional detailed screening of these 244 studies indicated that 35 contained data that were relevant to

support theory building and testing and hence were included. A further 12 papers were identified as relevant during the analysis through hand searching and expert consultation. At the end of the analysis, updating the searches identified 43 new abstracts of which 3 were included giving a total of 50 included papers (figure 2). A table summarising the included papers is given in online supplemental appendix 3.

Our results are structured by our programme theory. For each stage, we present a narrative explanation of our findings followed by a table of the CMOCs that underpin the explanation. We found that the evidence on some of the stages in our programme theory were clustered, and therefore, the following stages are presented together:

- ▶ 'Understanding abortion options' and 'Choosing a method/setting'.
- ▶ 'Accessing an abortion' and 'preabortion counselling assessing gestational age'.
- ▶ 'Taking mifepristone' and 'taking misoprostol'.

### Understanding abortion options and choosing a method

Choice of abortion methods and settings was influenced by service factors (number of appointments, timing and wait for appointments), personal responsibilities (caring/work commitments), geography (travel time/distance),

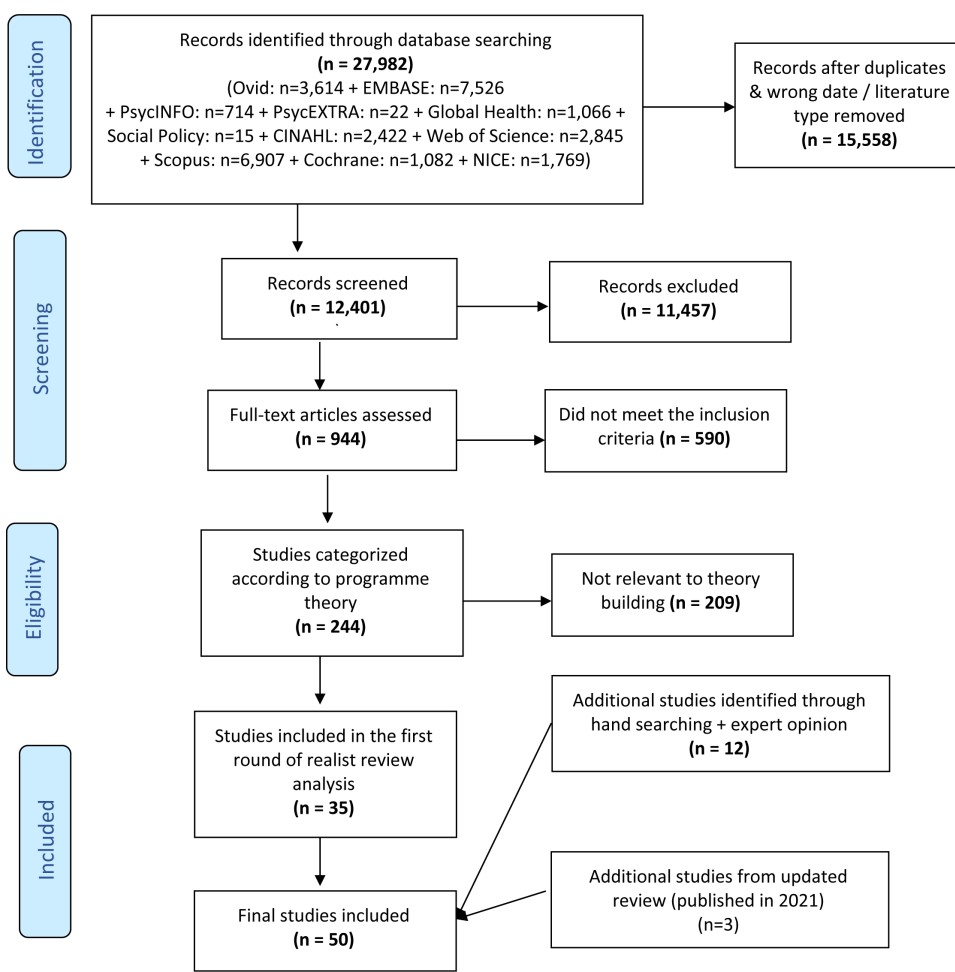

**Figure 2** PRISMA (The Preferred Reporting Items for Systematic Reviews and Meta-Analysis) diagram.

**Table 2** CMOCs for 'understanding options' and 'choosing a method'

| | CMOC |
|---|---|
| 1 | When people who want an abortion are offered a choice of procedures (c) they are more likely to be satisfied with their experience (ie, predict choice of this method again) (o) because they feel their decision is their own (ie, empowered in their choice) (m).[24 30 37–39 41 45 46] |
| 2 | Where the information provided about each procedure option includes clinical safety and effectiveness and experiential information (c) people are in a better position to make an informed choice (o) because they are more likely to have the information needed to do so (m)[42 76] |
| 3 | Where a range of personal experience is included that includes realistic accounts of pain/bleeding (c) then people are more likely to be satisfied with their choice (o) because they know what to expect and feel in control of the process (m)[32 40–44] |
| 4 | When people have social responsibilities in their lives (c), they will choose the method for abortion that fits best with these responsibilities (o) because of convenience (m)[21 30 32 77] |
| 5 | When people would prefer not to take an active role in the abortion process (c), they may choose a surgical abortion (o) because it enables them to distance themselves from the abortion process (m)[23 32 37 77 78] |
| 6 | When people have doubts about their abilities to carry out a medical abortion properly (c), they are more likely to request they are closely supervised (eg, complete the abortion within a healthcare context) (o), because of the emotional and practical support provided (m)[37–39] |
| 7 | When people have had what they consider to be an unpleasant experience of medical abortion (c), they will be unlikely to choose medical abortion again (o), because they want to avoid the experience (m)[24 38–40 43] |

CMOC, context-mechanism-outcome-configuration.

relationships and need for privacy/secrecy and the extent to which people wished to be aware of/involved in/in control of the process (table 2).[12 30–36] Those choosing medical abortion at homemade this decision on the basis of their social responsibilities (such as the need to provide childcare) or when a home abortion enabled them to maintain their confidentiality, rather than their personal preferences about the procedure itself.[12 21 30 36–38] Conversely, some people chose hospital care because it limits knowledge of the procedure among household members (eg, an abusive or unsupportive partner) and requires less responsibility for the process.[33 35] Although we were not specifically seeking literature on reasons for surgical abortion, people cited a wish to be asleep/unaware of the procedure or to avoid pain/bleeding/sight of the products of conception ortaking the mifepristone[37 39] or because it could be completed quickly.[40]

People valued frank descriptions from health professionals about the range of experience of medical abortion.[32 40–44] A participant in one study commented that 'It's good to do it at home but I think a good level of honesty from the nurse is helpful because it's on the cusp of being copeable with'.[39] Fear about the abortion procedure was the most frequently cited barrier to accessing an abortion in one study, with information as the most highly ranked facilitator to care.[42] High levels of satisfaction were associated with information on abortion delivered through short films in clinical contexts.[44–46] It is notable that in this study the film was rated as more informative than a leaflet even though the content was the same. Developing high-quality information resources is challenging because of the need to offer both clear clinical instructions and a wide range of experience that enables people to prepare themselves, their supporters and their surroundings. One study evaluating an animated account of one person's experience of medical abortion at home showed that this approach was well received but participants were concerned that their own experience was different from that shown.[44]

## Accessing a service/preabortion consultation/assessing gestational age

People value telephone/telemedicine self-referral options,[10 12 21] because of the privacy, autonomy and convenience they potentially afford and because they remove risk of judgmental attitudes from referring professionals (table 3). People rated distance to clinic and waiting times as more important than whether counselling was via telemedicine or face to face[31] and reasons for preferring a telemedicine consultation included the ability to engage from a more comfortable space (at home) and feeling more able to ask questions.[47] However, some people need to take action to speak privately, for example, completing the consultation in their car.[47 48]

The preabortion consultation is variable and may include an initial in-person consultation with/without an ultrasound scan and/or an examination or a fully remote service including self-reporting of gestational age.[22] Medical abortion without a prior ultrasound scan is safe[12] and in one qualitative study showed that some people preferred not to have an ultrasound scan because they associated it with having positive experiences of desired pregnancy[47] and in another study ultrasound was rated as unimportant for 53% of participants.

## Taking the abortion medications

Most of the literature focuses on the experience of taking misoprostol because this generates the most physical side

**Table 3** CMOCs for 'accessing a service/preabortion consultation'

| | CMOC |
|---|---|
| 1 | Where people can self-refer for abortion services via telephone or telemedicine services (c) they are more likely to be satisfied with this process (o) because they can avoid feeling judged by referring professionals (m).[22 48 79 80] |
| 2 | When people who want privacy, convenience and rapid access to abortion are offered the initial consultation by telephone or telemedicine (c), they are more likely to be satisfied (o), because their remote consultations are often available more quickly and are less likely to disrupt existing responsibilities (m)[21 30 31 47 77] |

CMOC, context-mechanism-outcome-configuration.

effects of medical abortion; however, some studies suggest that people may find taking the mifepristone the most emotional part of the treatment regimen and that this should be acknowledged in the information and support provided (table 4).[41] Self-administering misoprostol enabled people to control the timing of their abortion and prepare by making a comfortable space or obtaining things that they might need such as analgesia or hot water bottles, fit the abortion around other responsibilities or keep it private.[24] When given a choice of how to take the misoprostol, 70% in one study chose to self-administer it sublingually, rather than vaginally because they felt more confident that they had inserted it correctly using this approach.[21] Side effects from the misoprostol were common, with 77% of people experiencing one of the common symptoms (nausea, vomiting, diarrhoea or headaches).[21]

## Managing expulsion

Experience of managing the expulsion of the products of conception is highly variable (table 5). People interviewed after medical abortion at home point to the importance of understanding 'what is normal' as an important factor to help manage anxiety and to guide seeking clinical advice. Remote clinical support is important and highly valued and is used by 10%–50% of people who choose medical abortion at home.[21 24 38 49]

While most people find the process acceptable, the evidence shows that some experience pain and bleeding that is unexpected or distressing and cite this as the reason for future preference for an alternative method.[24 38 39] In one study, 94% of respondents had experienced pain with a median reported pain score of 6.7 (on a scale of 1–10) though 57% reported their pain as better or the same as they were expecting.[50]

Ninery-three per cent took analgesia. This is consistent with mean maximum pain scale of 6–8 reported elsewhere.[51] In one study, half of people experienced more bleeding than they anticipated[38] and in another 77% of participants reported that it was more than their usual menstrual period.[50]

The pain relief offered to people completing medical abortion at home is not standard across services. When people completing a medical abortion at home were provided with a combination of paracetamol, ibuprofen and rectal diclofenac, almost all used all three medications, one quarter accessed additional analgesia and 65% used additional non-pharmacological measures such as heat and rest to reduce pain.[38] In recent UK studies, 93% used pain relief (41%–92% paracetamol, 39%–61% ibuprofen 64% dihydrocodeine/codein).[25 50]

Most people who experienced significant pain and bleeding felt more comfortable coping with this at home, '…it is such a physical and emotional process so, y'know, at home's better. You basically need to make friends with your toilet for 8 hours, so at home's better. I was such a mess. I would be embarrassed for people to see me like that.'.[39] However, a minority would have preferred to have in-person health professional care because of the practical and emotional support provided.[23 24 38]

When people having medical abortion at home have had a previous pregnancy (ending in live birth, abortion or miscarriage), they are less likely to report pain that is distressing or to request professional support in comparison with those who have not had a previous pregnancy. However, they are no less likely to report heavy bleeding.[38] One reason to choose medical abortion at home is that people wish to be aware of the process of the abortion. Some find it helpful to know that the pregnancy has

**Table 4** CMOCs for 'taking the abortion medication'

| | CMOC |
|---|---|
| 1 | When people who want an abortion have reservations about their active participation in the process they may prefer not to self-administer the medications (o) because they feel personally responsible for ending the pregnancy (m)[41] |
| 2 | When people are given the option of self-administering misoprostol (c) they are better able to prepare for the process (o) because they can choose the time and place of the medical abortion (m)[24] |
| 3 | When people lack confidence in their ability to self-administer misoprostol (c) they prefer that this be done by a health professional (o) because they can then be confident that the procedure is done correctly (m)[21 24 39] |

CMOC, context-mechanism-outcome-configuration.

**Table 5** CMOCs for 'managing expulsion'

| | CMOC |
|---|---|
| 1 | Where people experience pain or bleeding that is outside the limits of what they were told to expect (c) they are more likely to request clinical support (o) because they are concerned that their health may be at risk (m)[23 24 38] |
| 2 | Where people having medical abortion at home have also had a previous pregnancy (live birth/abortion/miscarriage) (c), they are less likely to report pain that is distressing and to request professional support (o) because the experience is more familiar (m)[38 48] |
| 3 | Where people receive adequate pain relief during medical abortion (c) they are more likely to be satisfied with the whole experience of medical abortion at home (o) as they were more comfortable (m)[24 38 39] |
| 4 | Where people want to have privacy when having an abortion (c), they may choose to manage the abortion process at home (o) because they feel less exposed (m)[36 39] |
| 5 | Where people have family/friends/partners whom they know will be supportive when they have an abortion (c) then they value their presence during a medical abortion at home (o) because of the practical and emotional help that they provide (m).[24 52] |

CMOC, context-mechanism-outcome-configuration.

been passed if they see the products of conception, while others find it distressing.[39]

Some people having an abortion valued the support of partners, family and friends for company or practical support while others valued the opportunity to complete the abortion alone.[24] Partners (all men in cited study) interviewed about their experience of supporting people having a medical abortion expressed feelings of sadness, worry and concern, and sometimes ambivalence, about the decision to have an abortion as well as valuing the opportunity to support their partners through the process.[52]

### Assessing the success of abortion

There is good evidence to suggest that people can self-assess the success of abortion[53–58] (table 6). There was little evidence in the literature of people's thoughts and feelings about this process.

### Choosing contraception

For some people the time of abortion provision is a convenient, appropriate and valued opportunity to discuss contraception and national/international guidelines recommend this (table 7).[54] For those people opting for implants, having them inserted when medical abortion medication is provided, or for those choosing an intrauetine devices, an appointment booked shortly after medical abortion, means they are more likely to receive their contraceptive of choice.[39 59 60] However, medical

**Table 6** CMOC for 'assessing the success of abortion'

| | CMOC |
|---|---|
| 1 | When people are made aware of the symptoms and signs that indicate that the abortion has been successful (c), they are able to make accurate assessments (o) because they know what to look for (m)[53–58] |

CMOC, context-mechanism-outcome-configuration.

abortion via telemedicine limits the opportunity to do this. For other people, a contraceptive discussion at the time of the medical abortion consultation is unwelcome[61] and may be associated with a sense of being judged and pressurised about contraception.[39] Some people also prefer to focus on managing their abortion before considering contraception.[39 61–63] In this situation the offer of a delayed telephone or face-to-face contraception discussion can be welcome, including for those who have had difficulty finding a contraceptive method that suits them.[39 64–67]

### DISCUSSION
### Choice of abortion procedure remains essential

Increased expectations of self-care, access to digital health services and the COVID 19 pandemic, may have reduced choice of abortion method. High levels of satisfaction are reported after medical abortion but variation in experience means that choice of method remains important.[21 23 32 37] Though medical abortion at home is a lower cost procedure than medical abortion in hospital which, in turn, is a lower cost procedure than surgical abortion,[68] personal preference, past experience or context may make some abortion processes, for some people, difficult, distressing or unsafe.[69 70] People who may have lower levels of satisfaction with medical abortion at home include those who: have not had a previous pregnancy, do not have access to a comfortable and private space at home, for example, those experiencing homelessness, do not have support from family/partners/friends, have inflexible personal or employment responsibilities, or wish to disengage from the process.

Realistic information on the range of experience of abortion at home is important so that the abortion does not generate surprise or anxiety and so that people can adequately prepare for it at home.

Informed decisions require information on the potential advantages and disadvantages of the available

**Table 7** CMOCs for 'choosing contraception'

| | CMOC |
|---|---|
| 1 | At the time of abortion provision (c) some people are happy to discuss or be provided with contraception (o) because of the salience of this issue for them and the convenience (m)[39] |
| 2 | When people opting for long-acting reversible contraceptives have an appointment booked for them when receiving abortion medication at a time and place of their choosing after the abortion, (c) they are more likely to receive their contraception of their choice (o) because of salience (m) and less work for users in making the appointment (m)[59 60] |
| 3 | When people who are focused on the abortion procedure are offered contraceptive advice at a later date (c), they are more likely to welcome a discussion about contraception (o), because they have the capacity to consider this (m)[39 64–67] |
| 4 | When people who have had a previous abortion are offered telephone contraceptive counselling following abortion (c) they are more likely to take this up (o) because they may want help with complex contraceptive needs (m)[64 66 67] |
| 5 | When contraception is raised at the time of abortion (c), people can feel judged (o) if healthcare providers imply that it is important to prevent future unintended pregnancies (m)[39] |
| C6 | When blended (digital linked to face to face) post abortion support and contraception services are designed with user input (c) they are likely to be acceptable and accessible to users (o) because they address their needs (m).[81] |

CMOC, context-mechanism-outcome-configuration.

treatment options and abortion providers should offer information on the effectiveness, safety, health risks and benefits and the expected range of experiences of all abortion options. Although the effectiveness of abortion procedures is important to inform choice as up to 5% may require a surgical evacuation to complete the abortion,[71–73] effectiveness was not mentioned in user accounts of reasons for choosing medical abortion at home. Our review shows that it is important to communicate: a range of experiences, the importance of context (eg, where there is lack of privacy) or personal experience (eg, those with previous pregnancies report less pain). There is limited recent research on the best strategies to communicate this information and further work is required to collate and test good practice.

### Acknowledging the implications of moving a process from a healthcare setting to home

Guidelines on support for medical abortion at home may not sufficiently acknowledge the work that people do to prepare their space, manage their privacy, fulfil their work and caring obligations around the process, make decisions on pain relief, clean household areas, decide when and how to take misoprostol, assess their experience against what is normal and decide when to ask for professional help. There is a need to understand the new types of information/advice required to support people who do this work. The response may involve new technologies (eg, medical abortion support at home apps), new or developing roles (eg, medical abortion doulas) or new forms of peer support (eg, discussion forums to help with preparation and share experience). Further research is required to understand and define health service responsibilities when procedures move from clinically facilitated to self-managed in general and for medical abortion at home in particular.

### Understanding optimal pain relief

There is no consensus within the literature on analgesia for those who experience pain during medical abortion although a Cochrane Review is currently in progress on this topic.[74] There is currently significant variation in practice described in terms of the analgesics offered, the schedule and route of administration and whether these are provided by health services.

### More choice within the procedure

Enabling people to tailor their abortion experience could be an empowering strategy within a process that is often experienced as disempowering.[33 75] Options include when and how to take medications, pharmaceutical and other strategies for pain relief, the amount and form of support available and when contraception is discussed.

### Contraception

Periabortion contraception counselling should be delivered without pressure to take up contraception as some people find this stigmatising. A choice of when and where to discuss contraception is important. The option of delayed counselling/provision is helpful for some and may be the only option for those who are having intrauterine devices or implants and choose telemedicine abortion.

### Strengths and limitations

We were systematic and transparent in our approach to the realist review which was conducted in accordance with the RAMSES standards.[27] Our authorship team represents a variety of clinical and academic backgrounds, ensuring divergence in our analysis and we benefited from expert feedback from the multidisciplinary SACHA team throughout the review. Limitations include our analysis on publicly accessible literature, located through recognised research databases and Google. We focused on the settings where abortion is legal to draw conclusions

most relevant to the UK context, which meant our eligibility criteria screened out settings where abortion is not currently legal, which often included low-income and middle-income country studies. There were clear gaps in the evidence that we found and we have highlighted these in our conclusions.

### Further research needed

Based on our review of the literature, we recommend that further research is needed on:

► Optimal strategies for providing information and supporting choice of procedure/setting.
► Understanding how medical abortion at home involves or impacts family and friends (beyond partners).
► Optimal medications for pain relief including the dosing, route of administration and length of time that pain relief is required.
► How to offer choice within the abortion procedure in a manner that is empowering and not overwhelming
► Optimal means for providing contraception advice/support after the abortion procedure.
► Perspectives and experiences of healthcare providers: as part of the SACHA study, we are conducting a scoping review on healthcare practitioner preparedness to provide abortion globally, and a quantitative survey of practitioners in Britain, aimed at determining their experiences and attitudes towards abortion provision.

### Recommendations

Our recommendations are summarised in our revised programme theory in figure 3 .

► People should be given a choice of type and place of abortion.
► Discussion of choice of method should include service factors, personal responsibilities, geographical context and preferred level of awareness of/engagement with the procedure.
► Information to support choice of procedure should include the range of experience of medical abortion and realistic accounts of pain and bleeding.
► Health system support for the abortion process should acknowledge the new types of work that people take on when they manage this process at home, for example, preparing a space for the abortion.
► There should be opportunities to personalise experience including when and where to take medications, who is present, where to pass products of conception.
► Support for taking mifepristone should acknowledge the emotional significance of this for some people.
► Adequate pain relief is an essential part of satisfaction with medical abortion, particularly for those who are pregnant for the first time. Pain relief should be offered as well as information on non-pharmacological strategies to manage pain.

**Figure 3** Revised programme theory with key learning from this review.

► People should be offered a choice of where and when to discuss contraception including opportunities to book long-acting reversible contraceptives appointments if this is their preference.

## CONCLUSIONS

Acknowledging the work done by patients when moving an intervention from clinic to home is important. This includes preparing a space, managing privacy, managing work/caring obligations, deciding when/how to take medications, understanding what is normal, assessing experience and deciding when and how to ask for help. Strategies that reduce surprise or anxiety and enable preparation and a sense of control support the transition of this complex intervention outside healthcare environments. It is important that options for those who are unable to take on these additional responsibilities or who prefer not to do so are maintained, including clinical facilities and appropriately trained clinicians who can provide surgical abortion and medical abortion within healthcare facilities.

**Collaborators** The SACHA study team: Sharon Cameron, Louise Keogh, Patricia Lohr, Jennifer Reiter, Sally Sheldon, Jill Shawe, Annette Aronsson.

**Contributors** RF, KW, CF and PB conceptualised the study. GW was responsible for the methodology. RF, KW, CF, WVN, PB, GW, RM, MJP, RS and ML wrote the protocol. ML registered the review in the Prospero system (ID CRD42021225307) and conducted the searches. All authors screened the papers. WVN conducted preliminary data extraction. All authors participated in the data analysis. AI composed the tables. PB, CF, WVN and GW wrote the first draft. All authors contributed to the final manuscript. PB is the guarantor. This paper reports on data collected as part of the SACHA Study: Shaping Abortion for Change, funded by the National Institute of Health Research. (www.lshtm.ac.uk/sacha) Collaborators of the SACHA Study provided feedback on all stages of the study.

**Funding** This paper is a part of the SACHA Study, funded by the National Institute of Health Research, award NIHR129529.

**Competing interests** None declared.

**Patient and public involvement** Patients and/or the public were involved in the design, or conduct, or reporting, or dissemination plans of this research. Refer to the Methods section for further details.

**Patient consent for publication** Not applicable.

**Provenance and peer review** Not commissioned; externally peer reviewed.

**Data availability statement** Data are available on reasonable request. This is a review and the articles are available from appropriate journals.

**ORCID iDs**
Wendy V Norman http://orcid.org/0000-0003-4340-7882
Maria Lewandowska http://orcid.org/0000-0002-3012-1132
Rachel Scott http://orcid.org/0000-0003-0304-823X

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
