## [Reviewer comments · BMJ Open]

ARTICLE DETAILS

TITLE (PROVISIONAL)	Improving experience of medical abortion at home in a changing therapeutic, technological and regulatory landscape: a realist review
AUTHORS	Baraitser, Paula; Free, Caroline; Norman, Wendy; Lewandowska, Maria; Meiksin, Rebecca; Palmer, Melissa; Scott, Rachel; French, Rebecca; Wellings, Kaye; Ivory, Alice; Wong, Geoff

VERSION 1 – REVIEW

REVIEWER	Schaaf, Marta Independent Consultant
REVIEW RETURNED	15-Aug-2022

GENERAL COMMENTS	Congratulations to the authors for a clear, straightforward paper on an important topic. The CMOCs and programme theory are clear. I have very few comments to share (the fewest I have ever had). - The first sentence in the section entitled "Searching for evidence and document selection" is not clear. It was only after reading subsequent sections that I understood what that sentence meant.- The word "homeless" is used as a risk factor in the discussion section. To err on the side of being clear about why, it might be good to list this next to the statement about having a private space at home, ie "do not have access to a comfortable and private space at home or are homeless." I have no other comments to make.
--

REVIEWER	Walsh, Aisling Royal College of Surgeons in Ireland, Public Health and Epidemiology
REVIEW RETURNED	19-Aug-2022

GENERAL COMMENTS	This is an important and well conducted/written review that will greatly add to the knowledge of home medical abortion. Some minor revisions suggested here. - Given that one objective is focused on the user perspective – but indeed the entire review seems to be more focused on the perspective of the user - this could be reflected in the title and aim. A review from the perspective of the service providers would also yield important information to inform UK service development.- "Our findings relate to settings where abortion is legal". How was that determined through the search?- Abstract: "Peer-reviewed literature from 01/01/2000 to 09/12/2021, describing interventions or models of abortion care." Should read interventions or models of home abortion care?
---

	Methods: - Justification for starting the search from 2000. Results - What about differences in different contexts and cultures? The importance of context is mentioned, but only locally, for example a lack of privacy. Context from a country perspective would also be very important and needs to be discussed throughout the results. - Linked to the above point, the studies included in the review are from a limited number of national/geographical contexts: many from Scotland, USA, UK more broadly, Germany, Canada. It should explicitly be stated that the evidence is from a small number of countries/contexts. Also, a lack of lower and middle income country research, apart from Cambodia. - The revised programme theory seems to be based on advice to health system/staff: 'offer' and 'provide'. This should be acknowledged that this the review focuses on the user perspective, and that research into perspectives of what health professionals should 'offer' and 'provide' would also be important. Further research - Perspectives of health professionals.
--	--

VERSION 1 – AUTHOR RESPONSE

Reviewer: 1

Dr. Marta Schaaf, Independent Consultant

Comments to the Author:

Congratulations to the authors for a clear, straightforward paper on an important topic. The CMOCs and programme theory are clear. I have very few comments to share (the fewest I have ever had).

- The first sentence in the section entitled "Searching for evidence and document selection" is not clear. It was only after reading subsequent sections that I understood what that sentence meant.

We added an explanation further up in the methods:

"In the literature review component of the study, we sought to investigate novel models of abortion care addressing current therapeutic, technological, and regulatory trends, which would be relevant to the UK in the next 5 years. We subsequently divided this work between two reviews: a scoping review discussing the healthcare practitioner and system preparedness for abortion provision, which is currently in progress, and this realist review, focused on improving the experience of medical abortion at home."

We also rephrased the paragraph under 'searching for evidence and document selection' to:

"Our evidence search strategy started with a comprehensive literature search designed to answer a broader question for the SACHA Study: 'Interventions of models of abortion care/provision: addressing current therapeutic, technological and regulatory trends which would be relevant to or feasible in the UK in the next 5 years.' This search elicited the full set of articles which were subsequently used to inform both this realist review and a scoping review."

- The word "homeless" is used as a risk factor in the discussion section. To err on the side of being clear about why, it might be good to list this next to the statement about having a private space at home, ie "do not have access to a comfortable and private space at home or are homeless."

We changed this to read:

“People who may have lower levels of satisfaction with medical abortion at home include those who: have not had a previous pregnancy, do not have access to a comfortable and private space at home, e.g., those experiencing homelessness, do not have support from family/partners/friends, have inflexible personal or employment responsibilities, or wish to disengage from the process.”

I have no other comments to make.

Reviewer: 2

Dr. Aisling Walsh, Royal College of Surgeons in Ireland

Comments to the Author:

This is an important and well conducted/written review that will greatly add to the knowledge of home medical abortion. Some minor revisions suggested here.

- Given that one objective is focused on the user perspective – but indeed the entire review seems to be more focused on the perspective of the user - this could be reflected in the title and aim. A review from the perspective of the service providers would also yield important information to inform UK service development.

The aim of this review was for it to be about users' experience, but the data we have drawn on is not necessarily only from the user perspective - a lot of these experiences were reported through studies carried out with providers. This is because in our screening we specifically didn't exclude study participants who were not service users. Hence whilst our findings are about user experiences, the data used to inform this are much broader. Hence we did not feel it is appropriate to revise the title as it may mislead readers into thinking that we have only drawn on a narrower range of data to inform our findings.

- “Our findings relate to settings where abortion is legal”. How was that determined through the search?

It was determined through screening using the WHO database.

World Health Organization. Global Abortion Policies Database: United Nations; 2018 [cited 2019 Mar 25]. Available from: <https://abortion-policies.srhr.org/>.

We added the following sentence to the methods: “We screened the papers using the WHO database (<https://abortion-policies.srhr.org/>) to include only studies reporting on research conducted in settings where abortion is legal. “

- Abstract: “Peer-reviewed literature from 01/01/2000 to 09/12/2021, describing interventions or models of abortion care.” Should read interventions or models of home abortion care?

Changed as suggested.

Methods:

- Justification for starting the search from 2000.

Although a small handful of countries initially introduced mifepristone medical abortion in the early 1990's, general usage of mifepristone became much more prevalent after 2000. Further, from the year 2000 we note a range of relevant advances in health service delivery and technology. Based on the content expertise of the SACHA project team we judged it was reasonable and sufficient to look at health services research published within the past 20 years to inform our exploration of novel approaches to care.

Results

- What about differences in different contexts and cultures? The importance of context is mentioned,

but only locally, for example a lack of privacy. Context from a country perspective would also be very important and needs to be discussed throughout the results.

We agree that contexts will vary across different settings, people and countries. We also accept that it would be almost impossible to develop fine-grained detailed explanations (in the form of context-mechanism-outcome-configurations (CMOCs)) for each and every possible contextual variation. It is because of this challenge that we chose to use a realist review approach. To briefly explain, the purpose of a realist review is not to exhaustively explain all phenomena in all contexts. Instead, it seeks to identify common patterns of outcomes across settings (e.g. countries) or people and explain when and why these may occur (through the use of CMOCs). The warrant for the transferability of these findings is based on a realist assumption about the nature of the causal processes (i.e. mechanisms) – which is that they are inherent ‘liabilities’ of people or objects that are widely present but not necessarily ‘activated’ unless relevant contexts are present. A focus on mechanisms does mean that any CMOCs we produce to explain outcomes does tend to be at a higher level of abstraction [Evidence-Based Policy: A Realist Perspective. Pawson D. 2006, Sage, London.]. It also means that we have had to conceptualise context in broader and more inclusive ways [<https://www.tandfonline.com/doi/full/10.1080/13645579.2021.1918484>]. We thus believe that the CMOCs we have developed are likely to provide explanations that are transferable and applicable in many countries where abortions are legal.

- Linked to the above point, the studies included in the review are from a limited number of national/geographical contexts: many from Scotland, USA, UK more broadly, Germany, Canada. It should explicitly be stated that the evidence is from a small number of countries/contexts. Also, a lack of lower and middle income country research, apart from Cambodia.

We focused on data that was useful for implementation in the UK, and consequently were looking for contexts where abortion was legal – which unfortunately also does eliminate data from many low- and middle-income countries. Further, we comprehensively included studies that met this eligibility criteria, although acknowledge that due to local and regional limitations it may well be that research meeting our criteria was not published from smaller countries where abortion was legal. . We added this to our limitations that our evidence is focused on this context:

“We focused on the settings where abortion is legal to draw conclusions most relevant to the UK context, which meant our eligibility criteria screened out settings where abortion is not currently legal, which often included low- and middle-income country studies.”

- The revised programme theory seems to be based on advice to health system/staff: ‘offer’ and ‘provide’. This should be acknowledged that this the review focuses on the user perspective, and that research into perspectives of what health professionals should ‘offer’ and ‘provide’ would also be important.

As noted in our revisions above to our method section, an associated study currently being conducted by our team is a scoping review on this topic – we have added a bullet point on this to the ‘further research’ section: “• Perspectives and experiences of healthcare providers: as part of the SACHA Study, we are conducting a scoping review on healthcare practitioner preparedness to provide abortion globally, and a quantitative survey of practitioners in Britain, aimed at determining their experiences and attitudes towards abortion provision.”.

Further research

- Perspectives of health professionals.

Added.

VERSION 2 – REVIEW

REVIEWER	Walsh, Aisling Royal College of Surgeons in Ireland, Public Health and Epidemiology
REVIEW RETURNED	05-Oct-2022
GENERAL COMMENTS	My suggestions and commnets have been dealt with adequately. Thank you.